# Tactile-Guided Dynamic Contrastive Koopman Operator for Deformable Linear Object Manipulation

Aohua Liu[1], Kun Qian[1*], Boyi Duan[1], Shan Luo[2]

## I. INTRODUCTION

**D**EFORMABLE linear objects (DLOs) are commonly handled in tasks such as cable installation in smart construction [1] and hose routing in the manufacturing industry [2]. Similar to humans, robots are capable of following the contour of the in-hand DLO guided by high-resolution tactile perception [3] [4]. Such a contour following motion ensures the installation tension, thereby facilitating cables or hoses routing. Reliable robotic manipulation of DLO requires accurate modeling of how the robot's movements influence the DLO's in-hand pose. Obtaining a precise interaction model between the DLO and the moving gripper remains challenging for several reasons: 1) The contour following involves durable, nonlinear, and complex interaction dynamics. 2) The high dimensionality of the tactile feedback further complicates the task of accurately identifying the interaction model and achieving precise motion control.

The main contributions of this work are as follows:

1) A Dynamic Contrastive Koopman Operator (DCKO) is proposed to model the nonlinear, high-dimensional deformable interaction dynamics. By incorporating a task-oriented distance metric and a dynamic negative sampling-based contrastive learning strategy, the Koopman operator can extract task-relevant representations and equivalent distance metrics, thereby enhancing modeling interpretability.

2) A DCKO-based MPC controller (DCKMPC) is developed based on the embedded linear model, presenting improved tracking accuracy and computational efficiency.

3) The framework is validated on an in-hand DLO contour-following task and a high-level routing task, demonstrating enhanced prediction and control capabilities in high-dimensional servoing, along with its adaptability to more complex application scenarios.

## II. DYNAMIC CONTRASTIVE KOOPMAN OPERATOR

### A. Architecture of the DCKO model

The compliant interaction dynamics model between the manipulator's end-effector motion $\boldsymbol{u}_t$ and the in-hand DLO

*Corresponding author
[1]School of Automation, Southeast University and the Key Laboratory of Measurement and Control of CSE, Ministry of Education, Nanjing 210096, China. [2]Department of Engineering, King's College London, London, WC2R 2LS, United Kingdom. Email: kqian@seu.edu.cn. This work was supported in part by the Basic and Applied Basic Research Foundation of Guangdong Province (2025A1515010397) and the National Natural Science Foundation of China (No.61573101).

tactile state $\boldsymbol{o}_t$ is defined as the discrete-time nonlinear system:

$$\boldsymbol{o}_{t+1} = \boldsymbol{g}(\boldsymbol{o}_t, \boldsymbol{u}_t), \quad t \in \mathbb{N} \tag{1}$$

The Koopman theory [5] is well-known for its ability to linearly model nonlinear systems in an embedded space, facilitating the design of optimal controllers. The structure of the proposed DCKO is illustrated in Fig. 1. The encoder projects the tactile state $\boldsymbol{o}_t$ into the embedded space as $\boldsymbol{z}_t = \boldsymbol{\psi}(\boldsymbol{o}_t)$. The auxiliary network then generates the equivalent embedded control input $\boldsymbol{c}_t = \boldsymbol{h}(\boldsymbol{z}_t) \odot \boldsymbol{u}_t$, capturing the nonlinear components of the control input related to the state variables. The future state $\hat{\boldsymbol{z}}_{t+1}$ is then predicted by the linear time-invariant model in the embedded space:

$$\hat{\boldsymbol{z}}_{t+1} = \boldsymbol{A}\boldsymbol{z}_t + \boldsymbol{B}\boldsymbol{c}_t \tag{2}$$

Here, The Koopman operator $\boldsymbol{K} = [\boldsymbol{A}, \boldsymbol{B}]$ is composed of two learnable constant matrices.

### B. Dynamic contrastive learning strategy for DCKO

In the embedded space, contrastive learning optimizes $\boldsymbol{\psi}(\cdot)$, $\boldsymbol{h}(\cdot)$, and $\boldsymbol{K}$ by aligning positive samples and separating negative ones. As illustrated in Fig. 1, the predicted $\hat{\boldsymbol{z}}_{t+p}$ serves as the anchor, while its ground-truth counterpart $\boldsymbol{z}_{t+p} = \boldsymbol{\psi}(\boldsymbol{o}_{t+p})$ is regarded as the positive sample. To enhance the interpretability of the model and achieve better performance in downstream control tasks, we introduce a task-oriented distance metric for tactile states and employ it to allocate negative samples for each anchor.

**1) Definition of Task-oriented Distance Metric:** Since lighting, color, and noise in tactile images are irrelevant to deformable interaction dynamics, we employ a 2D parameterization technique to assess the observation equivalence between the tactile states $\boldsymbol{o}_i$ and $\boldsymbol{o}_j$, i.e.,

$$\|\mathcal{R}(\boldsymbol{o}_i) - \mathcal{R}(\boldsymbol{o}_j)\| := \beta \cdot |r_i - r_j| + |b_i - b_j|, \quad \beta \in (0,1) \tag{3}$$

Here, $\mathcal{R}(\boldsymbol{o}_i) := [b_i, r_i]$, where $b_i$ and $r_i$ denote, respectively, the position and the orientation of the contact area of $\boldsymbol{o}_i$ with respect to the tactile sensor's axis $\vec{x}$. Furthermore, to guide the model's focus on factors sensitive to dynamic behavior, a behavioral difference metric that quantifies the disparity in state responses under identical control inputs is defined as:

$$\mathcal{B}(\boldsymbol{o}_i, \boldsymbol{o}_j) := \max_{\boldsymbol{u} \in \mathcal{U}} \|\boldsymbol{D}(\boldsymbol{o}_i, \boldsymbol{u}) - \boldsymbol{D}(\boldsymbol{o}_j, \boldsymbol{u})\| \tag{4}$$

where $\mathcal{U}$ is the constraint set of feasible actions, and $\boldsymbol{D}(\boldsymbol{o}_i, \boldsymbol{u}) = \boldsymbol{A}\,\boldsymbol{\psi}(\boldsymbol{o}_i) + \boldsymbol{B}\,\boldsymbol{h}(\boldsymbol{\psi}(\boldsymbol{o}_i)) \odot \boldsymbol{u}$ represents the

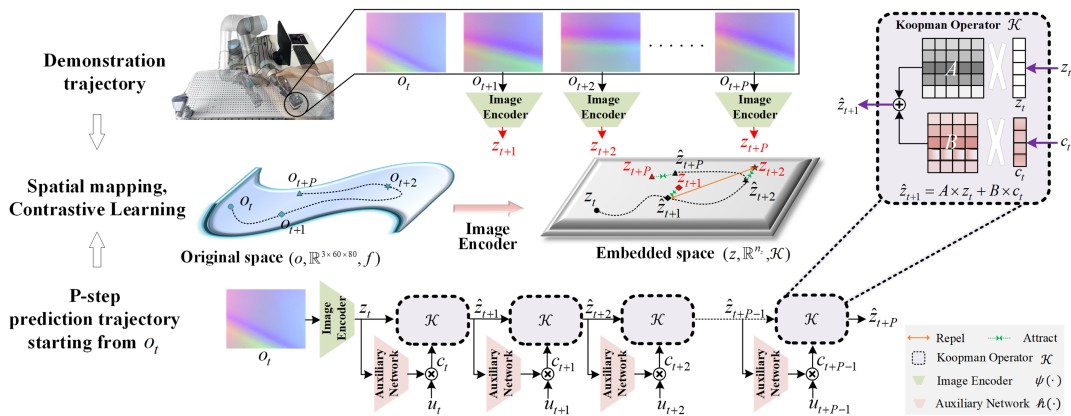

Fig. 1. Overview of the proposed DCKO model pipeline.

predicted dynamic response in the embedded space under $\boldsymbol{u}$. Finally, the *Task-oriented Distance Metric* is defined as:

$$d(\boldsymbol{o}_i, \boldsymbol{o}_j) := \gamma \cdot \mathcal{B}(\boldsymbol{o}_i, \boldsymbol{o}_j) + \|\mathcal{R}(\boldsymbol{o}_i) - \mathcal{R}(\boldsymbol{o}_j)\| \quad (5)$$

**2) Dynamic Negative Sampling based Prediction Contrastive Loss:** Using the *Task-oriented Distance Metric*, the $k$ farthest samples from each anchor form the negative set $\mathcal{D}_k^-$. The DCKO is trained with the following contrastive loss:

$$\mathcal{L}_{\text{DCKO}}(\theta_e, \theta_h, \boldsymbol{A}, \boldsymbol{B}) = \frac{1}{J(T-1)} \sum_{j=0}^{J-1} \sum_{t=0}^{T-2} \frac{1}{P(t)} \sum_{p=1}^{P(t)}$$

$$- \log \frac{S(\hat{\boldsymbol{z}}_{t+p}^j, \boldsymbol{z}_{t+p}^j)}{S(\hat{\boldsymbol{z}}_{t+p}^j, \boldsymbol{z}_{t+p}^j) + \sum_{i=1}^{k} \frac{d_i}{d} S(\hat{\boldsymbol{z}}_{t+p}^j, \boldsymbol{\psi}(\boldsymbol{o}_i)))} \quad (6)$$

where $\boldsymbol{o}_i \in \mathcal{D}_k^-$; $J$ is the batch size; $P(t) = \min(P, T-1-t)$ is the valid prediction horizon for a trajectory of length $T$; and $S(\boldsymbol{z}_i, \boldsymbol{z}_j) = \exp\left(-\|\boldsymbol{z}_i - \boldsymbol{z}_j\|^2\right)$ is a similarity function.

## III. DCKO-BASED MODEL PREDICTIVE CONTROL

To overcome the inevitable modeling errors of the Koopman-based dynamic model caused by the limited embedding dimension and external disturbances, the proposed robust controller $\boldsymbol{c}_t$ consists of two components:

$$\boldsymbol{c}_t = \boldsymbol{c}_t^{\text{KMPC}} + \boldsymbol{K}(\boldsymbol{z}_t - \bar{\boldsymbol{z}}_t) \quad (7)$$

Here, $\boldsymbol{c}_t^{\text{KMPC}}$ is acquired by solving the following receding optimization problem to reduce the tracking error $\bar{\boldsymbol{e}}_t$ between the nominal state $\bar{\boldsymbol{z}}_t$ and the desired state $\boldsymbol{z}_t^d$:

$$\min_{\bar{\boldsymbol{c}}_{t:t+H-1},\, \bar{\boldsymbol{z}}_{t+1:t+H}} \sum_{i=0}^{H-1} \left( \|\bar{\boldsymbol{e}}_{t+i+1}\|_{\boldsymbol{Q}^2} + \|\bar{\boldsymbol{c}}_{t+i}\|_{\boldsymbol{R}^2} \right) + \|\bar{\boldsymbol{e}}_{t+H}\|_{\boldsymbol{\Pi}^2}$$

$$\begin{aligned}
\text{s.t.} \quad & \boldsymbol{z}_t = \boldsymbol{\psi}(\boldsymbol{o}_t), \\
& \bar{\boldsymbol{e}}_{t+i+1} = \bar{\boldsymbol{z}}_{t+i+1} - \boldsymbol{z}_{t+i+1}^d, \\
& \bar{\boldsymbol{z}}_{t+i+1} = \boldsymbol{A}\bar{\boldsymbol{z}}_{t+i} + \boldsymbol{B}\bar{\boldsymbol{c}}_{t+i}, \\
& \bar{\boldsymbol{c}}_{t+i} \in \bar{\mathcal{C}}_{t+i}, i \in [0, \ldots, H-1]
\end{aligned} \quad (8)$$

The mismatch between the nominal state $\bar{\boldsymbol{z}}_t$ and the actual encoded state $\boldsymbol{z}_t$ is ultimately uniformly bounded when the feedback gain $\boldsymbol{K}$ satisfies the following Riccati equation:

$$2\boldsymbol{A}^\top \boldsymbol{F} \boldsymbol{A} + 2\mathbf{K}^\top \boldsymbol{B}^\top \boldsymbol{F} \boldsymbol{B} \mathbf{K} + \boldsymbol{A}^\top \boldsymbol{F} \boldsymbol{B} \mathbf{K}$$
$$+ \mathbf{K}^\top \boldsymbol{B}^\top \boldsymbol{F} \boldsymbol{A} + \sigma \mathbf{I} - \boldsymbol{F} = -\boldsymbol{G}. \quad (9)$$

where $\sigma > 0$, and $\boldsymbol{F}$ and $\boldsymbol{G}$ are positive definite matrices.

## IV. RESULTS AND CONCLUSION

The performance of our framework is evaluated against the state-of-the-art Koopman operator BKDDNN [5], with the experiment video available at https://youtu.be/DCKO.

**1) Model Identification Evaluation:** The modeling performance, measured by the Pearson correlation coefficient $r(d_e', d')$ between the original and embedded spaces, reaches 0.94 for ours versus 0.87 for BKDDNN, indicating superior modeling interpretability and prediction accuracy.

**2) DLO Contour Tracking Experiment:** We repeated the task ten times and evaluated performance with three indicators: the *per-step runtime* (17 ms vs. 37 ms), the *average tracking error* of the desired tactile state (0.01 vs. 0.10), and the *average follow rate* (100% vs. 95%), for our framework and BKDDNN, respectively. The proposed method outperforms the existing Koopman-based approach by aligning the distance metrics between the embedded and original spaces.

**3) Hose Routing Performance:** The routing task involves the challenges of approaching the designated fixed clips while preventing the hose from falling off midway. We repeated the task ten times and achieved a 90% success rate, demonstrating the effectiveness of our framework in handling complex tasks without retraining.

**Conclusion:** In this article, we proposed a concise and effective Koopman-based framework for the tactile servo control of the DLO. Through the application to the tactile servo task of the DLO in-hand contour following, the proposed approach demonstrates superior performance in both modeling and control compared to representative baselines. Furthermore, this algorithm can be directly compatible with more complex hose routing scenarios without any need for retraining.

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
