# OpenReview forum: "Tactile-Guided Dynamic Contrastive Koopman Operator for Deformable Linear Object Manipulation"
_IEEE.org/IROS/2025/Workshop/Tactile_Sensing — IROS 2025 Workshop Tactile Sensing OralPoster_

### Official Review · Reviewer_V8UA · 2025-09-23
**Need more detail description**

**Rating:** 7
**Confidence:** 5

**Review:**

In this paper, the authors first proposed a dynamic contrastive Koopman operator to model the dynamics of the DLP, and they futher developed a MPC controller based on it. The effectiveness of the method has been validated on an in-hand contour following task.

More technical details should be added to help the readers understand the work, such as the shape and exact defination of u_t, o_t, c_t, r_i, b_i. Visualization of them on Fig.1 can be added.

---

### Official Review · Reviewer_jfEf · 2025-09-23

**Rating:** 7
**Confidence:** 4

**Review:**

This paper presents a novel and well-motivated framework, the Dynamic Contrastive Koopman Operator (DCKO), for tactile-servo control of deformable linear objects (DLOs). The work is clearly significant, addressing the challenging problem of modeling high-dimensional, nonlinear tactile interaction dynamics for precise manipulation. The proposed integration of a task-oriented distance metric and dynamic negative sampling within a contrastive learning framework is a distinctive and original contribution, effectively enhancing the interpretability and relevance of the learned Koopman embedding for control tasks. The application to both contour-following and complex hose routing scenarios demonstrates strong practical utility and generalization capability.

The key strengths of the work include its superior performance over the baseline (BKDDNN) in prediction accuracy and control efficiency, as evidenced by the quantitative results, and its ability to transfer to more complex tasks without retraining. However, the paper would benefit from a more detailed discussion of the limitations. For instance, the robustness of the task-oriented metric to significant variations in tactile sensor characteristics or different DLO materials is not thoroughly evaluated. Additionally, while the real-world validation is commendable, the experimental scale remains relatively limited. The abstract also contains placeholder text that significantly detracts from the manuscript's professionalism and must be revised. Overall, this is a compelling contribution to tactile-based deformation control with a solid methodological foundation and promising results.